# Memory-Based Navigation in Elephants: Implications for Survival Strategies and Conservation

**DOI:** 10.3390/vetsci12040312

**Published:** 2025-03-30

**Authors:** Margot Morel, Robert Guldemond, Melissa A. de la Garza, Jaco Bakker

**Affiliations:** 1Broadway Veterinary Group, Unit 1 The Links, Herne CT6 7FE, UK; 2Conservation Ecology Research Unit, Department of Zoology and Entomology, University of Pretoria, cnr Lynnwood Road and Roper Street, Hatfield 0028, South Africa; robert.guldemond@up.ac.za; 3Michale E. Keeling Center for Comparative Medicine and Research, University of Texas MD Anderson Cancer Center, Bastrop, TX 78602, USA; made4@mdanderson.org; 4Animal Science Department, Biomedical Primate Research Centre, 2288 GJ Rijswijk, The Netherlands; bakker@bprc.nl

**Keywords:** elephant cognition, spatial memory, human–elephant coexistence, movement ecology, conservation planning, ecosystem engineers, climate resilience

## Abstract

Elephants exhibit exceptional memory capabilities, allowing them to adapt to environmental changes and human presence. Their ability to remember the locations of water, food, and safe pathways helps them survive in increasingly fragmented landscapes. Older elephants, especially matriarchs, play a key role in leading herds, yet how knowledge is transferred to younger generations remains uncertain. Conservation strategies can benefit from understanding elephant movement patterns by preserving migratory routes, ensuring access to essential resources, and mitigating human–elephant conflict. Protecting experienced individuals is vital, as their loss may disrupt herds and decrease survival chances. Elephants also shape ecosystems by dispersing seeds and maintaining habitat diversity, making them essential for biodiversity and climate resilience. By aligning conservation efforts with their natural behaviours, such as establishing wildlife corridors, reducing conflicts with farmers, and considering elephant decision-making in habitat planning, humans can foster coexistence while preserving these keystone species. Future research should focus on how elephants develop spatial knowledge and adapt to rapid environmental changes. Understanding their navigation strategies is key to ensuring their survival and maintaining the ecosystems they help sustain.

## 1. Introduction

Renowned for their remarkable intelligence and complex social structures, elephants (African forest elephant [*Loxodonta cyclotis*], African savannah elephant [*Loxodonta africana*], and Asian elephant [*Elephas maximus*] (Figure 1 and Figure 2) [1,2,3] exhibit notable cognitive abilities, enabling them to navigate and adapt to diverse and often challenging environments. Among these abilities, spatial and social cognition are essential for navigating their shrinking and changing habitats. Their advanced cognition and ecological impact make elephants keystone species, meaning their presence and behaviours influence the structure and function of ecosystems [4]. Navigation, the process by which elephants acquire, organise, and recall spatial information, is integral to their ability to locate resources, avoid threats, and adapt to environmental changes [5,6]. While it is likely that elephants rely on spatial memory or mental representations to navigate their environments, the exact mechanisms underlying this capacity remain unclear. This review addresses a critical research gap: the lack of empirical evidence determining whether elephants use structured internal maps, develop network-based spatial frameworks, or rely on habitual route memory. Our central research question is as follows: To what extent can current neuroanatomical and behavioural evidence reconcile with field observations to clarify the mechanisms driving elephant navigation and decision-making? We aim to synthesise insights from neuroscience, field ecology, and conservation studies to evaluate the validity of prevailing theories and highlight directions for future research. The findings indicate that elephants rely on memory-based navigation strategies; however, the exact nature of their spatial representation remains unclear. Some researchers propose that elephants form structured internal maps, mental representations of spatial relationships between landmarks, that allow for flexible navigation across unfamiliar terrain. Others suggest a network-based framework, where movement decisions are made based on learned associations between specific routes and destinations, without requiring an overall mental layout. A third hypothesis is habitual route memory, in which elephants rely on repeated exposure and familiarity, following well-established paths without forming a broader representation of the landscape. The hypothesis that elephants use structured internal maps to navigate remains speculative, as direct experimental validation in wild settings is limited and challenging to obtain.

Distinguishing between these mechanisms is essential for conservation planning, as it underscores the need to protect critical environmental features and movement pathways, including wildlife corridors and seasonal migration routes, which support elephants’ abilities to navigate and access resources effectively.

Elephants, particularly older individuals within herds, play a vital role as repositories of social and ecological knowledge. Matriarchs, often the most experienced group members, demonstrate leadership in decision-making, particularly in resource selection and navigation. However, theories regarding the mechanisms by which knowledge is transmitted within elephant societies remain primarily speculative. While it is widely assumed that older individuals pass on critical information to younger generations, direct empirical evidence for intergenerational knowledge transfer is limited [7]. This intricate interplay of navigation and memory equips elephants to cope with natural challenges, such as droughts and anthropogenic pressures, including habitat fragmentation and human–wildlife conflict [8,9].

Fishlock et al. suggest that African forest elephants exhibit social learning through resource-use traditions, particularly in site fidelity and movement patterns [10]. Their study on forest elephants highlights how specific resource sites, such as mineral licks and water sources, are used consistently across generations. However, the choice of these locations cannot be entirely explained by resource quality alone. Instead, these traditions likely arise from individual learning and social facilitation, wherein younger forest elephants observe and mimic the behaviours of more experienced conspecifics. This process does not necessarily imply deliberate teaching or structured knowledge transmission but rather a cultural inheritance shaped by long-term social interactions.

Despite the absence of definitive proof of direct knowledge transfer, the role of experienced elephants in guiding movement, accessing resources, and responding to threats underscores their importance in maintaining group cohesion and resilience. The loss of such individuals, mainly due to poaching and human–wildlife conflict, may have cascading effects on herd stability and survival, as younger elephants may lack access to the accumulated wisdom necessary for effective decision-making.

By refining our understanding of how elephants navigate and exploit their environment, conservation efforts can better account for the importance of social structure in maintaining population resilience. Future research should aim to disentangle the extent to which knowledge transfer occurs within elephant societies and how disruptions in these networks affect their ability to adapt to changing landscapes.

## 2. Navigation

### 2.1. Definition

Navigation refers to the ability of animals to move purposefully through their environment to reach specific destinations such as water sources, foraging sites, or shelter using various spatial and environmental cues [11,12]. It involves multiple cognitive and sensory mechanisms, including memory, landmark recognition, path integration, and environmental cues such as topography, olfaction, and social information. Unlike simple stimulus–response behaviours, navigation requires decision-making based on prior experiences and real-time environmental assessments, allowing animals to adapt to dynamic landscapes. Cognitive mapping is the mental process through which individuals acquire, organise, store, and recall spatial information about their surroundings, creating an internal representation or “map” of their environment [5,6]. This process enables efficient navigation, decision-making, and adaptation to dynamic and changing environments. Cognitive mapping was initially conceptualised by Tolman, who demonstrated that rats navigating a maze could use learned spatial information to find shortcuts or alternative routes when familiar paths were blocked, showcasing the flexibility of cognitive mapping. This finding underscored that navigation involves more than simple stimulus–response behaviours and relies on an internal mental map [5].

O’Keefe and Nadel further expanded on the neurological basis of cognitive mapping, demonstrating the role of the hippocampus in spatial navigation [6]. Their research showed that this brain structure is necessary for encoding, storing, and recalling spatial memories, allowing animals to form and use cognitive maps. Finally, Presotto et al. highlighted how savannah elephants in Botswana utilised environmental features such as riverbeds and hills as spatial markers, creating mental representations that allowed them to traverse extensive territories efficiently [8]. These findings indicate that elephants rely on memory-based navigation strategies; however, the exact nature of their spatial representation remains unclear. Unlike controlled experimental studies that have demonstrated cognitive mapping in species such as rats and birds, no direct empirical evidence currently confirms that elephants construct a true cognitive map. Instead, current observations in elephants, such as the repeated use of specific routes or targeted long-distance travel, can often be explained by associative memory or learned movement patterns reinforced through experience. While some behaviours are consistent with the use of internal spatial maps, the absence of controlled studies limits our ability to conclusively distinguish between cognitive mapping and other forms of memory-based navigation. Therefore, elephants’ navigational behaviour should be interpreted cautiously, acknowledging both the plausible hypotheses and the lack of direct experimental validation. While elephants’ navigational behaviours, such as revisiting distant, temporally variable resources and following established routes, are consistent with cognitive mapping, it is important to emphasise that direct experimental validation in elephants is still lacking. Therefore, although there is converging evidence from neuroanatomical, spatial ecological, and behavioural studies suggesting the presence of advanced spatial memory, it remains more accurate to interpret elephant navigation as memory driven rather than definitively map based. This nuanced distinction reflects the current limitations of empirical data while acknowledging that multiple lines of indirect evidence support the plausibility of cognitive mapping in elephants.

Given the lack of direct evidence supporting cognitive mapping in elephants, their navigational behaviour is more accurately described as a memory-driven movement rather than the use of a defined mental representation of space. Distinguishing between these mechanisms is essential for conservation planning, as it underscores the need to protect critical environmental features and movement pathways, including wildlife corridors and seasonal migration routes, that support elephants’ ability to navigate and access resources effectively.

### 2.2. Neuroanatomical Basis

With an average brain mass of 4700 g, adult elephants have the largest brains among terrestrial animals [13]. This is roughly 13 times the average brain mass of chimpanzees (350 g) and 3.4 times that of humans (1400 g) [13,14]. Elephants’ exceptional spatial memory is linked to the complex neural architecture of their brains, especially the hippocampus, a key structure for spatial navigation and memory processing. The hippocampus in elephants is notably large and well developed, supporting the encoding, storage, and retrieval of spatial information required for navigating extensive and ecologically diverse landscapes. This structural complexity is comparable to that of other highly cognitive mammals, such as primates and cetaceans, who also demonstrate sophisticated spatial navigation abilities. In elephants, the hippocampus has been linked to the formation of detailed cognitive maps, allowing them to remember critical resources’ locations over large distances and long periods [15,16]. This region is densely packed with neurons and exhibits significant plasticity, which allows for the adaptation of neural connections based on new spatial experiences. The hippocampus in elephants shows a high density of pyramidal neurons, particularly in the CA1 and CA3 subfields, which are essential for spatial learning and memory consolidation. These neurons form part of a network that allows elephants to integrate various sensory inputs and create a detailed cognitive map of their surroundings [15,16]. The dentate gyrus, another region within the hippocampus, plays a fundamental role in pattern separation, distinguishing between similar spatial environments or objects. This is vital for elephants, given the overlapping landscapes they traverse. While most evidence of hippocampal function in elephants is anatomical or histological, some behavioural studies provide indirect support for its role in navigation. For instance, Presotto et al. and Polansky et al. used GPS tracking data to demonstrate that elephants consistently travel along memorised routes and return to specific resource sites across seasons [8,9]. These spatial behaviours imply hippocampal involvement, although no current studies have directly correlated neural activity (e.g., via neuroimaging or electrophysiology) with real-time navigation. Such interdisciplinary studies, combining neurobiology with behavioural ecology, remain a critical gap in elephant research. However, the observed capacity to recall distant, temporally variable resources suggests that the hippocampus plays a central role in storing and retrieving spatial information in real-world contexts. Additionally, the entorhinal cortex, which interfaces between the hippocampus and neocortex, contributes to spatial awareness by generating grid-like neuronal activity that encodes position and direction. This grid cell system allows elephants to form a stable internal representation of their position within a landscape, enabling efficient movement over large distances without reliance on external cues [17]. Neurogenesis, the process of generating new neurons, has also been observed in the hippocampal region of elephants. Evidence for adult neurogenesis in the dentate gyrus of elephants was demonstrated through immunostaining for doublecortin (DCX). A moderate density of DCX-immunoreactive cells was observed within the sub-granular zone and granule layer, with dendrites extending into the inner molecular layer of the dentate gyrus. This distribution aligns with neurogenesis patterns commonly seen in other mammals [18]. Finally, in conjunction with reduced cortical neuron density, there is typically an increase in overall neuron size. Haug reported that elephants have a mean neuron size of 4200 µm^3^, with the size distribution skewed towards large and very large neurons, primarily pyramidal cells. The neuron size in the elephant cortex surpassed that of all species measured by Haug, apart from one cetacean species [19]. Neurogenesis in adult mammals is closely linked to cognitive plasticity, particularly in brain regions such as the hippocampus. While direct evidence for adult neurogenesis in elephants is lacking, comparative studies suggest that similar mechanisms may be present. For example, research on large-brained mammals has revealed populations of prenatally generated neurons that retain immature molecular characteristics into adulthood, especially in the cerebral cortex and limbic regions [20]. Additionally, Amrein and Lipp highlight that adult hippocampal neurogenesis varies across species in relation to ecological and evolutionary traits, supporting the hypothesis that neurogenic strategies are shaped by life history demands [21].

Furthermore, comparative neuroanatomy studies suggest that elephants exhibit a high degree of cortical folding, or gyrification, in the brain, which increases the surface area and potentially supports advanced cognitive abilities. The elephant’s brain gyrification index is comparable to that of qualified, highly intelligent species such as humans and cetaceans, which may underlie their sophisticated problem-solving, social, and long-term memory capabilities [22]. This expanded cortical surface facilitates complex associative processing, enabling elephants to integrate diverse sensory information and contextual cues when navigating. The somatosensory cortex is highly developed and connected to the trunk. It allows elephants to utilise tactile exploration with spatial memory, further enriching their environmental understanding and navigation [22]. In addition to anatomical adaptations for spatial memory, the neurological health of elephants is a critical area of concern within veterinary science. While systematic studies on neurodegenerative diseases in elephants are limited, certain conditions have been documented that may impair neural function. For instance, floppy trunk syndrome (FTS) is a condition observed in savannah elephants characterised by trunk paralysis due to peripheral nerve degeneration, leading to difficulties in feeding and hydration [23]. Although the direct impact of FTS on spatial memory has not been established, the associated neural degeneration could potentially affect broader neurological functions. Additionally, captivity has been associated with neural changes in elephants, including reduced hippocampal activity and behavioural abnormalities, likely due to environmental deprivation and chronic stress [24]. These findings underscore the importance of environmental enrichment and appropriate social structures in maintaining neural health. Furthermore, studies on other species have shown that neurodegenerative conditions can lead to impairments in spatial memory and navigation. For example, in a mouse model of neurodegeneration, hippocampal neurons exhibited disrupted firing sequences, undermining spatial memory codes [25]. Understanding these mechanisms could provide insights for preserving cognitive functions related to navigation and spatial memory in aging elephants. Continued research into elephant-specific neurological conditions and their effects on cognitive mapping is essential for improving both veterinary care and conservation strategies.

## 3. Navigating the Landscape: The Use of Natural and Anthropogenic Landmarks in Elephant Movement

Elephants develop and rely on highly detailed information processing, which allows them to create and store intricate mental representations of landscapes, resource locations, and spatial relationships. These maps are invaluable for guiding elephants through expansive home ranges spanning hundreds of square kilometres [26,27].

### 3.1. Ecological Knowledge

Presotto et al. provided evidence that savannah elephants use prominent geographical features like hills, riverbeds, and mountains as reliable spatial markers. For instance, savannah elephants in Botswana were observed using these natural landmarks to guide their seasonal migrations across the Okavango Delta. Despite environmental changes, the consistency of their routes highlights the integration of these features into their navigation processes [8]. Such natural markers provide stable reference points in otherwise dynamic or featureless environments, helping elephants optimise their movements.

Elephants also use changes in vegetation as cues for navigation. Loarie et al. demonstrated that savannah elephants in Botswana’s Chobe National Park (NP) track seasonal vegetation changes to locate water sources and forage [28]. For instance, elephants returning to areas of acacia blooms during the wet season illustrate how vegetation types act as temporal landmarks tied to resource availability. This type of landmark recognition is essential for survival in environments where resource availability fluctuates significantly over time.

In arid regions like Namibia’s Etosha NP, savannah elephants navigate using vegetation gradients created by piosphere effects around waterholes. Polansky et al. observed that savannah elephants initiated targeted movements to waterholes from distances beyond the reach of sensory cues like smell or sight, suggesting reliance on memory and visual landmarks associated with these waterholes [9].

Elephants adjust their navigation strategies based on how familiar they are with their environment. In the central areas of their home range, they tend to navigate directly to locations using an Euclidean-like mapping approach. However, when moving into less familiar peripheral areas, they shift to using more established, habitual routes, demonstrating flexibility in their spatial memory. This behaviour is similar to findings in smaller mammals, such as rats, which have been shown to encode location information differently based on familiarity [9].

### 3.2. Ecological Applications

Elephants are often celebrated for their exceptional navigation skills, attributed to their ability to recall and utilise spatial and environmental information. While there is no definitive scientific evidence of cognitive mapping as a mental process in elephants, their behaviours strongly suggest the presence of advanced spatial memory and decision-making strategies. Tolman defines cognitive mapping as creating an internal mental representation of an environment, allowing an individual to navigate efficiently [5]. However, direct evidence of mental maps remains elusive in the case of elephants due to the challenges of studying cognitive processes in the wild. Despite the lack of direct proof, various field studies have provided compelling behavioural evidence suggesting that elephants rely on memory-driven navigation.

In arid environments such as Namibia’s Etosha NP, savannah elephants demonstrated remarkable long-distance navigation to access waterholes. Polansky studied elephants navigating to water sources up to 50 km away, bypassing closer but less reliable waterholes. The study concluded that these movements were not random but memory driven, with elephants recalling specific locations and conditions of waterholes based on past experiences [9]. For example, elephants frequently made direct, rapid movements to selected waterholes during dry seasons, even when vegetation around these sources showed no visual cues indicating water presence. This behaviour suggests that elephants rely on spatial memory rather than sensory information like smell or sight. Such targeted navigation reflects the sophisticated decision-making processes that allow elephants to maximise survival in extreme environments.

The Gourma elephants of Mali undertake one of the longest elephant migrations ever recorded, covering over 600 km annually through the semi-arid Sahel. This migration is driven by extreme seasonal variability in resource availability. During the dry season, these elephants travel long distances to reach reliable water sources such as Lake Banzena and return to grazing areas during the wet season [29]. What is particularly notable about the Gourma elephants is their ability to recall and navigate across landscapes that appear featureless to human observers. These migrations are not random; elephants consistently follow established routes that maximise resource access while minimising energy expenditure. The use of these routes over generations suggests a transfer of spatial knowledge. This behaviour aligns with the hypothesis that elephants use detailed cognitive maps to navigate challenging environments.

In Hwange NP, Zimbabwe, savannah elephants rely on artificial water points established by park authorities to sustain populations during the dry season. These waterholes are in predictable areas, and GPS tracking studies have shown that elephants navigate directly to these artificial sources year after year [30]. Elephants’ behaviour at Hwange demonstrates their ability to adapt to human-altered landscapes by incorporating new resources into their spatial memory. For example, when new waterholes were introduced, elephants quickly incorporated these into their movement patterns, indicating the flexibility and adaptability of their spatial memory. Additionally, elephants often prioritised visiting these waterholes over searching for natural sources, reducing the risks associated with exploratory behaviour in resource-scarce habitats. In Kenya, elephants changed their traditional migratory routes to avoid areas with high human activity. For instance, elephants in the Amboseli-Tsavo ecosystem adapted their movements seasonally, avoiding farms and villages where conflicts are likely to occur [31]. This behavioural flexibility indicated that elephants can modify their spatial strategies based on risk assessment. In Botswana’s Chobe NP, savannah elephants demonstrated seasonal shifts in movement patterns to access resources. During the wet season, elephants dispersed widely, but as water sources dried up, they converged on the Chobe River, following memory-driven migratory pathways likely passed down through generations [7].

## 4. The Role of Elders

### 4.1. Relational Mapping and Memory

Despite the absence of definitive proof of direct knowledge transfer, a growing number of behavioural studies suggest its occurrence in elephants. McComb et al. conducted playback experiments demonstrating that older matriarchs respond more accurately to social and environmental cues, indicating their accumulated knowledge and ability to guide herd responses [7]. It was observed that calves often follow matriarchs and older females during movements to water sources and feeding areas, learning these locations through repeated exposure. A controlled observational study documented how juvenile elephants altered their movement and foraging behaviours over time after repeated exposure to experienced individuals, implying learning through observation and experience [32]. While these studies do not involve experimental manipulations in the strictest sense, they offer strong field-based evidence that younger elephants acquire spatial knowledge and decision-making cues through social learning mechanisms. More controlled longitudinal studies would be valuable to explicitly test the extent and mode of knowledge transfer. These matriarchs responded to unfamiliar calls with increased alertness and protective behaviours, a skill that benefits the herd by helping them recognise potential threats or opportunities to fortify alliances. The researchers found that herds with older matriarchs who had a greater ability to remember social connections were better at avoiding conflicts and navigating complex societal landscapes, leading to higher overall reproductive success and group survival. The study concluded that matriarchal memory is a community repository, enabling elephants to maintain and recall complex social relationships crucial for herd stability and cohesion. This results underscored the evolutionary advantages of advanced memory and relational mapping in elephants by demonstrating how matriarchs act as repositories of social knowledge.

Elephants can recall past relationships and interactions, facilitating societal cohesion, enhancing stability, and enabling cooperation within herds. This factor is increasingly essential in habitats where resource competition is high [31]. A study by Bates et al. emphasised how savannah elephants use relational memory to form a mental network of individual relationships and locations, allowing them to make complex, memory-based decisions about social interactions and territory [33].

Researchers played back the calls of different family members, such as a close relative or a distantly related family member, from a hidden location. The researchers found that the elephants responded differently depending on the location from which the call was played. For example, if a close family member’s call came from an unexpected direction, the elephants showed visible signs of vigilance, often moving towards the sound source or signalling alarm. The study revealed that elephants maintain a mental map of their family members’ usual positions and movements, adjusting their behaviours based on this relational mapping [34]. The accuracy with which elephants could track family members’ locations and respond to unexpected placements demonstrated that they possess complex cognitive abilities for heightened spatial awareness. This mapping ability is thought to help elephants coordinate movements across large areas and sustain cohesive family groups, which is critical for survival and social functioning in the wild.

It was demonstrated that family groups led by older matriarchs (over 60 years old) responded more appropriately to predatory threats than groups with younger matriarchs (under 35 years), selecting defensive behaviours correctly in 85% of playback trials with unfamiliar lion roars compared to just 50% in younger-matriarch-led groups [35]. This superior decision-making capacity directly correlates with enhanced reproductive success and calf survival within the herd. However, aging in elephants, while not yet fully characterised, may still lead to vascular and inflammatory brain changes. Chysud et al. identified signs of microvascular changes in aged elephants, including glial scarring and neuronal shrinkage in hippocampal tissue samples collected post-mortem from 14 individuals over 40 years of age, which may compromise spatial memory [36].

Human-induced trauma also has direct consequences on navigation and survival. Kock reported 16 cases of flaccid trunk syndrome in free-ranging elephants in Zimbabwe, where affected animals showed bilateral trunk paralysis and progressive weight loss due to feeding impairment [23]. These cases were fatal in over 50% of instances. Snare injuries, another common consequence of human–elephant conflict, account for up to 30% of elephant trauma cases treated by field veterinary teams in east Africa annually, with frequent complications such as osteomyelitis or sepsis requiring surgical debridement or limb amputation [37]. Such physical impairments reduce mobility, disrupt migratory routines, and may force affected elephants to take more dangerous or conflict-prone routes in search of resources. In addition, psychological trauma and chronic stress are well-documented in both orphaned and translocated elephants. Bradshaw et al. described that orphaned juvenile elephants translocated without mature social leaders exhibited erratic migration, disrupted social behaviours, and hyperaggression [38]. These individuals were responsible for 90% of fatal attacks on rhinos in Pilanesberg NPbetween 1994 and 2000, a phenomenon attributed to impaired neurological development and social dysregulation. Elevated faecal cortisol concentrations, rising up to 400% above baseline within 48 h of translocation, have been recorded in elephants in Kenya and South Africa, remaining elevated for up to 6 weeks after release [39]. These physiological stress responses correlate with increased pacing, disorientation, and diminished foraging efficiency.

Moreover, translocation often fails to achieve its intended conservation goals. Fernando et al. tracked 12 male elephants translocated in Sri Lanka; 10 of them (83%) returned to their original home ranges within 30 days. During this process, three individuals were killed in renewed human–elephant conflict incidents [37]. This highlights not only the strength of elephants’ spatial memory but also the unintended consequences of interrupting their cognitive map and social context.

Veterinary teams therefore play a critical role in assessing not only physical trauma but also the cognitive and neurobehavioural well-being of elephants affected by conflict or management interventions. Protecting the functional cognitive capacities of elder elephants is vital, as their decline may impact not only individual survival but also herd-level navigation and resilience in fragmented landscapes.

### 4.2. Experienced Elders in Elephant Navigation, Social Learning, and Communication

The matriarch is pivotal in the elephant group, providing essential leadership and extensive knowledge of resource locations while coordinating collective defence efforts [40,41,42]. Her dominant rank not only influences the hierarchical standing of other females within her kinship group but also shapes patterns of group resource utilisation [43]. Moreover, families led by older matriarchs demonstrate a superior ability to discern conspecific vocalisations and adjust their responses based on the familiarity of the caller [7]. The presence of a dominant matriarch often prompts distinct rumble vocalisations among group members [44]. Groups with older matriarchs also exhibit heightened exploratory behaviour in response to unfamiliar vocalisations, a trait less pronounced in groups with younger matriarchs [29]. Thus, matriarchs are essential in enriching social bonds and relationships with other herds or families [45]. Studies indicate that older matriarchs, with longer lifespans and more extensive knowledge, lead herds more effectively regarding resource acquisition and survival strategies [31].

Older matriarchs in savannah elephant herds, particularly those studied in Kenya’s Amboseli NP, have been shown to play a crucial role in herd survival by relying on accumulated social knowledge [7]. Herds led by older matriarchs showed stronger, more immediate responses to potential dangers, highlighting the importance of the matriarch’s social memory for herd cohesion and survival. This study underscores the conservation value of protecting older matriarchs, as their loss could leave herds more vulnerable to threats, including human–elephant conflict (HEC). Experienced matriarchs can help navigate and avoid human-dominated areas, suggesting that their role is vital in natural and human-influenced environments.

The intergenerational transmission of knowledge, particularly cognitive maps, is an essential mechanism by which elephants pass on critical spatial knowledge and decision routes. In Kafue NP, Zambia, savannah elephants rely on the leadership of matriarchs to navigate the park’s complex floodplain system. Seasonal flooding transforms the landscape, rendering previously accessible areas impassable while opening new resource-rich regions. Younger elephants learn to adapt to these shifting conditions by observing the matriarch’s routes, prioritising safe crossings and avoiding areas prone to rapid water level changes [46]. A study by Chamaillé-Jammes et al. found that elephants with experienced matriarchs were more likely to traverse these floodplains successfully and locate dry grazing areas, highlighting the role of matriarchal leadership in survival under seasonal pressures. Following decades of civil war in Mozambique, many elephants in Gorongosa NP lost traditional migratory routes due to significantly reduced matriarchal populations [30]. As elephant numbers recovered, younger generations began relearning these routes through the guidance of surviving older females. Stalmans et al. documented how matriarchs led herds to rediscover seasonal waterholes and mineral deposits that had not been used for decades, demonstrating the resilience and adaptability of matriarchal knowledge even after periods of disruption [47]. Elephants adjust their navigation strategies based on how familiar they are with their environment. GPS tracking data from Presotto et al. showed that in the central areas of their home range, savannah elephants navigated directly to resource locations, suggesting a map-like spatial understanding. In contrast, movements in peripheral or less familiar areas followed more established, habitual routes [8]. These observations are based on direct tracking data and support the hypothesis that elephants switch between different memory systems depending on environmental familiarity, similar to findings in smaller mammals such as rats [9].

Much of elephants’ survival knowledge is passed down through direct observation. This form of learning, where younger elephants observe and replicate the behaviour of older, more experienced herd members, plays a central role in teaching critical skills such as navigating migratory routes, identifying resource locations, and responding to environmental threats. For instance, young elephants learn migratory routes by following the matriarch along established paths. By following the matriarch over time, younger elephants learn movement patterns and develop their spatial knowledge, likely internalising key resources and routes used by the herd [48].

In India’s Nagarhole NP, Vidya et al. examined how Asian elephants navigate landscapes increasingly shaped by human activity, mainly focusing on their ability to avoid high-risk areas such as roads and villages [49]. Through field observations and spatial analysis, the study revealed that older, more experienced elephants, including matriarchs, were fundamental in guiding herds away from areas of potential conflict. Asian elephants were observed altering their movement patterns based on the density of human presence, avoiding roads during peak traffic hours and bypassing villages entirely, regardless of the time of day. Younger elephants appeared to learn these behaviours by following and observing the actions of older herd members, gradually incorporating this risk-avoidance knowledge into their spatial awareness and movement patterns. The study also noted the flexibility of Asian elephants’ spatial strategies, as herds often detoured significantly to avoid high-risk zones, demonstrating both foresight and adaptability. The loss of experienced individuals, such as matriarchs, could severely disrupt this intergenerational knowledge transfer, increasing herd vulnerability to human-induced risks and highlighting the need for conservation efforts to mitigate HEC and preserve matriarchal structures.

A study by McComb et al. investigated how savannah elephants use vocal cues to assess group familiarity, age, and gender, which aids in determining threats. In Amboseli NP, Kenya, the researchers used playback experiments to observe how herds responded to recordings of different vocalisations: contact calls from other elephants, threatening calls from lions, and human voices of different ethnic groups [22]. Elephants showed stronger reactions to unfamiliar calls, particularly from adult males, suggesting they could recognise potential threats by voice alone. Herds led by older matriarchs demonstrated quicker, more intense reactions, including defensive groupings, indicating the matriarch’s role in interpreting vocal cues and teaching younger elephants threat responses. This study highlights how matriarchal experience in recognising vocal cues is essential for herd survival. This study also underscores the role of the matriarch in teaching the herd how to respond to vocal cues, especially those that signal potential danger. Through repeated exposure to the matriarch’s responses, younger elephants learn to associate specific sounds with certain actions, such as grouping defensively or remaining alert. These learned behaviours become part of the herd’s collective knowledge, essential for long-term survival, especially in areas where human–elephant interactions or predator threats are prevalent.

## 5. Adapting to Habitat Change

Elephants face increasing challenges from climate change, including altered vegetation, water scarcity, and habitat fragmentation. However, their advanced cognitive and behavioural flexibility enables them to adapt through mechanisms such as navigation and dietary flexibility. HEC poses significant challenges to elephant survival, with climate change and human expansion exacerbating these issues. Elephants may exhibit remarkable cognitive flexibility and behavioural adaptations to navigate conflict zones, often altering their behaviour and movement patterns to mitigate risks [50,51].

### 5.1. Altered Vegetation, Water Availability, and Habitat Fragmentation

Climate change is reshaping ecosystems, leading to shifts in the distribution and availability of vegetation and water resources. Water is an essential resource for elephants, with individuals consuming up to 200 litres daily [52,53]. Climate-induced changes in rainfall patterns and prolonged droughts are shrinking natural water sources, creating severe challenges for elephant survival.

For instance, Foley et al. examined the influence of gender, maternal experience, and family group characteristics on calf survival in a savannah elephant population during a severe drought in Tanzania’s Tarangire NP in 1993 [54]. Their findings revealed that young male calves were particularly vulnerable to the drought, and calf mortality was higher among inexperienced mothers than among experienced ones. Additionally, significant variation in calf mortality was observed across family groups, with those remaining within the NP experiencing greater losses than groups that migrated outside the park.

Another report from Loarie et al. investigated how savannah elephants adapt their seasonal vegetation preferences in a semi-arid savannah, focusing on the interplay between resource availability, water proximity, and landscape features [55]. It highlights the critical role of dietary flexibility and water dependency in coping with environmental variability exacerbated by modelled climate change. The study combined satellite imagery (NDVI) and field observations to analyse vegetation quality and availability across seasons in relation to satellite-tracked elephant movements and proximity to water sources. Statistical models evaluated the influence of seasonal rainfall, vegetation patterns, and water availability on habitat use. The findings revealed that elephants preferred nutrient-rich grasslands during wet seasons but shifted to woody vegetation near permanent water sources in the dry season. This reliance highlights their dietary flexibility and vulnerability to climate-induced changes, as erratic rainfall and prolonged droughts limit access to critical resources, increasing stress and competition. In Hwange NP, Zimbabwe, Chamaillé-Jammes et al. illustrated a decline in palatable grasses due to increased temperatures and reduced soil moisture, resulting in an over-reliance on less nutritious forage by savannah elephants, such as mopane leaves and bark. Elephants often use plant phenology, such as acacia blooms, as a cue for resource availability [30].

Mali’s Gourma elephants rely on seasonal wetlands and rivers for drinking water and grazing. Wall et al. found that these wetlands are drying earlier each year, forcing elephants to migrate longer distances between water points [29]. This increased travel increases energy expenditure and the risk of human–wildlife conflict. In Botswana’s Okavango Delta, decreased rainfall reduces water flow in perennial rivers, impacting the region’s capacity to sustain large elephant populations. Elephants increasingly rely on artificial waterholes, which can lead to overcrowding and resource depletion [56]. In Namibia’s Etosha NP, savannah elephants are increasingly forced to share waterholes with livestock due to the drying of natural sources. The study by Huang et al. used satellite collar tracking data from elephants across southern Africa and combined t with remote sensing and landscape analysis to model movement corridors and habitat connectivity [57]. The methodology included identifying key migration routes, mapping water sources, and assessing how elephants navigate fragmented landscapes influenced by human settlements and agricultural activities. The study found that elephants adapt their movements to utilise remaining corridors but often share waterholes and grazing lands with livestock, particularly during dry seasons. This overlap may increase competition for resources and exacerbate HEC. The research emphasises the importance of preserving migratory pathways and strategically managing shared resources to mitigate conflict and support elephant conservation.

A study by Viljoen examines the habitat selection and resource use of desert-dwelling savannah elephants in the northern Namib Desert in Namibia, highlighting their remarkable adaptability to water-scarce environments [58]. By analysing movement patterns and feeding behaviours, the research shows that these elephants travel extensive distances between water sources, often over 60 km, to access limited water supplies during dry periods. They rely heavily on deep wells they dig in dry riverbeds and their memory of seasonal waterholes. Additionally, their diet shifts to include drought-tolerant vegetation such as mopane trees and other woody plants, especially when grasses and other preferred forage are unavailable. These adaptive behaviours reduce competition with other herbivores and optimise resource use in an arid ecosystem, showcasing elephants’ capacity to adjust their movements and feeding strategies in response to extreme environmental conditions. In Botswana’s Chobe NP, savannah elephants have incorporated artificial waterholes into their ranging system, timing visits based on human refilling activities [59]. These examples showcase how elephants integrate environmental and anthropogenic cues into their navigation strategies, ensuring survival in increasingly water-stressed landscapes.

One study used aerial surveys, camera traps, and historical data to assess the impact of Mozambique’s civil war on wildlife in Gorongosa NP and to examine post-conflict recovery patterns [60]. Aerial surveys provided population estimates and distribution data for large mammals, including elephants, focusing on habitats near water sources and regions heavily affected by war. Camera traps were strategically placed to monitor elephant behaviour and movement, capturing evidence of their use of safer, less-disturbed areas and rediscovered historical migration routes. By comparing post-war data to historical records, the researchers found that elephants exhibited behavioural adaptations to habitat fragmentation and human activity, including avoiding mined areas and shifting their range to less accessible terrain. These findings highlight the species’ resilience and adaptability in navigating fragmented and disrupted habitats. In India’s Rajaji NP, Asian elephants have adapted to increasing habitat fragmentation caused by expanding human settlements and infrastructure, such as roads and railways. Joshi et al. conducted a study using telemetry and direct observations to track elephant movements and their strategies for navigating fragmented landscapes [61]. The study revealed that elephants increasingly rely on narrow forest corridors and cross railway tracks at night to minimise human interactions. They also modify their migratory routes to include agricultural fields, where they forage on crops, compensating for reduced access to natural food sources.

A 2016 study presented the findings of the Great Elephant Census (GEC), the first standardised, continent-wide survey of savannah elephant populations [62]. The GEC estimated approximately 352,271 elephants across 18 countries, covering about 93% of the species’ range in those nations. The data revealed a significant decline, with populations decreasing by an estimated 144,000 elephants between 2007 and 2014, equating to an 8% annual reduction, primarily due to poaching. Notably, 84% of the remaining elephants were found within protected areas; however, many of these areas exhibited high carcass ratios, indicating elevated mortality rates. Habitat fragmentation exacerbates these declines by disrupting elephants’ migratory routes and access to essential resources, leading to increased HEC and heightened vulnerability to poaching. Fragmented habitats hinder elephants’ ability to adapt to climate change, as they struggle to find water and forage during extreme weather events, further threatening their survival. The study underscores the urgent need for comprehensive conservation strategies that address both poaching and habitat fragmentation to halt the downward trajectory of savannah elephant populations [62].

### 5.2. Navigating Human–Elephant Conflict

HEC is a growing concern driven by habitat encroachment, agricultural expansion, and resource competition. Elephants often adapt their behaviours to exploit human agricultural resources, such as maise, bananas, and rice, leading to economic losses and retaliatory actions.

In the Laikipia-Samburu Ecosystem, Kenya, Graham et al. conducted a comprehensive spatial analysis of savannah elephant movements, using satellite data to map conflict zones [63]. Farmers have successfully reduced crop-raiding incidents in Sri Lanka’s Mahaweli region by installing beehive fences. A study by King et al. reported that beehive fences reduced crop damage by elephants by up to 85%, with elephants avoiding farms protected by active hives [64]. Similarly, in Kenya, beehive fences showed a deterrence success rate of 80% in trials conducted over two crop seasons [65]. Elephants integrate these deterrents into their spatial memory, altering movement patterns to bypass protected fields.

This approach not only safeguards crops but also promotes coexistence by respecting elephants’ natural behaviours and aversions. Another quantitative study in Tanzania revealed that farms with beehive fences experienced significantly fewer elephant incursions compared to unprotected control farms (average of 1.3 vs. 6.5 crop-raiding events per month) [66]. These results demonstrate that the memory-based avoidance of aversive stimuli, like bee sounds, can be a reliable strategy in mitigating conflict. Combined with community engagement and local maintenance, such methods offer an ecologically sound and cost-effective solution to reduce HEC.

Crop-raiding incidents were concentrated within 3 km of forest edges, where agricultural expansion overlapped with elephant corridors. Farmer interviews and crop damage surveys quantified economic losses. The findings highlighted how elephants adjusted their behaviours based on seasonal crop availability. In addition, elephants display remarkable ingenuity in overcoming deterrent measures. In Sri Lanka’s Mahaweli region, Asian elephants timed their crop-raiding activities to nighttime hours, reducing the likelihood of human encounters [67]. Elephants showcase remarkable adaptability by evaluating and adjusting to their surroundings, seamlessly integrating human-induced challenges into their spatial problem-solving strategies. Social learning plays a key role in this process, as younger herd members closely observe and replicate the successful techniques of older, more experienced individuals.

Sukumar analysed conflict incidents over 15 years, finding that over 70% occurred in regions where paddy fields and coconut plantations bordered Asian elephant habitats [68]. The elephants selectively targeted high-yield crops during the harvest season, indicating their ability to identify and exploit human resources. Conflict mapping was combined with dietary analysis using dung samples to determine crop consumption. Satellite imagery was used to track and correlate habitat changes with conflict hotspots. In Assam, India, HEC is a persistent issue, with crop raiding being one of the most significant challenges. Asian elephants frequently enter agricultural fields to feed on rice, bananas, and sugarcane, especially during harvest seasons, behaviours driven by habitat loss and fragmentation. Another study examined the relationship between habitat loss and the frequency of HEC in Assam [69]. The authors found that as forest cover decreases below a critical threshold, crop-raiding incidents and property damage by elephants increase significantly. The study highlighted the importance of maintaining adequate forest habitats to mitigate conflict and suggested that conservation efforts should focus on preventing further habitat fragmentation.

Infrastructure such as highways and railways intersecting migratory routes poses significant risks to elephants, yet they exhibit advanced adaptations to mitigate these dangers. In Kenya’s Tsavo ecosystem, savannah elephants have been observed utilising underpasses and culverts constructed beneath the Standard Gauge Railway (SGR) to move safely between habitats. A study by Okita-Ouma et al. monitored elephants and found that 78% of crossings through these underpasses occurred at night, indicating a behavioural adaptation to minimise human interaction and reduce the risk of accidents [70].

These examples (Table 1) collectively highlight elephants’ reliance on spatial memory and learned movement patterns to navigate and adapt to environmental and anthropogenic challenges. In water-scarce environments like the Namib Desert, elephants use detailed spatial memory to recall distant seasonal waterholes and self-dug wells, enabling survival over broad, arid terrains. Similarly, in Botswana’s Chobe NP, elephants incorporate human activities, such as refilling artificial waterholes, into their navigation strategies, demonstrating their ability to integrate dynamic environmental cues into mental representations. In fragmented habitats like Mozambique’s Gorongosa NP, elephants have adapted their migration routes by rediscovering historical paths and avoiding mined areas, showcasing their ability to adjust spatial knowledge based on risk. Likewise, in India’s Rajaji NP, elephants navigate narrow forest corridors, time railway crossings at night, and alter routes to include agricultural fields, all strategies facilitated by mental maps that minimise conflict and ensure resource access. Even when coping with agricultural expansion and infrastructure, as seen in Sri Lanka and Kenya, elephants assess deterrent measures like electric fences or utilise railway underpasses, reflecting their capacity to evaluate and adapt to environmental changes through learned and shared spatial knowledge.

Veterinary interventions have become increasingly critical in mitigating the physiological and ecological consequences of habitat change for elephants. Free-ranging elephants navigating fragmented landscapes or drought-prone ecosystems are at heightened risk of disease, injury, and malnutrition. In particular, elephant endotheliotropic herpes virus (EEHV), which causes acute haemorrhagic disease in juveniles, has emerged as a leading cause of death in captive and wild elephants. In a comprehensive review, Long et al. reported a case fatality rate of up to 85% for Asian elephant calves infected with EEHV-1A, with most deaths occurring within 48 h of clinical onset [71]. Detection in wild-born calves in India and Myanmar has expanded concern beyond zoo-managed populations, prompting calls for field-deployable qPCR testing and plasma therapy access in veterinary field units. Early intervention, often dependent on veterinarians identifying signs like oral cyanosis or facial oedema, can improve survival, with successful treatment documented when antiviral medication (famciclovir or acyclovir) and fluid support are administered promptly.

Moreover, gastrointestinal parasitism, especially strongyle-type nematodes, can compromise immunity and worsen nutritional stress in drought-affected populations. In a faecal egg count study in southern Africa, strongyle burdens in elephants increased by 42% during dry seasons compared to wet periods, correlating with lower forage availability and body condition scores below 3/5 [72]. Veterinarians conducting routine monitoring and deworming in conservation areas like Hwange and Kruger NP help mitigate such subclinical burdens, reducing long-term morbidity.

To better understand how elephants respond to landscape fragmentation, veterinarians and conservation biologists have collaborated on GPS collaring initiatives that also support medical surveillance. In Kenya, Gara et al. deployed GPS collars on adult elephants in Amboseli NP and surrounding dispersal zones to assess the influence of vegetation productivity (measured using NDVI) and human infrastructure on habitat use [73]. Elephants showed strong seasonal fidelity to wet-season ranges (NDVI > 0.35) but shifted towards high-conflict zones during dry months, coinciding with lower NDVI values (<0.20) and increasing the risk of injury, crop raiding, or poaching. Collar data showed that elephants in dry months travelled 20–35 km more per day on average, with resting time reduced by 12%, suggesting heightened energetic and physiological strain.

Together, these veterinary contributions, ranging from disease diagnosis to data-informed collaring, enhance our understanding of how elephants cope with environmental uncertainty and underscore the importance of integrating wildlife health monitoring into conservation planning.

## 6. Implications for Elephant Conservation, Biodiversity, and Human Communities

Elephants’ spatial memory and navigational abilities underpin their survival and adaptation to dynamic environments, making them an essential consideration in conservation efforts. Beyond safeguarding elephant populations, these abilities have far-reaching implications for biodiversity preservation and human communities.

### 6.1. Ecological Benefits of Elephant Migration

Elephants’ navigation abilities are critical in their function as ecosystem engineers. Their capacity to recall and navigate to specific resources enables behaviours that sustain ecological balance. In Namibia’s Etosha NP, savannah elephants may have used spatial memory to rotate their visits to different waterholes, preventing the overuse and degradation of any single site [62]. This deliberate distribution, guided by spatial memory and learned movement patterns, stands to minimise their ecological footprint, support habitat recovery, and ensure the availability of resources for other species reliant on these water sources.

Elephants’ ability to navigate long distances underpins their capacity for migration, which is vital for maintaining genetic diversity and connecting fragmented habitats. In Central Africa’s Congo Basin, forest elephants rely on spatial memory to traverse vast landscapes, enabling them to disperse seeds of different species, many of which depend exclusively on these movements for regeneration [6]. The continuity of essential ecosystem services is preserved by safeguarding these migratory corridors, demonstrating how elephants’ navigational abilities support biodiversity and forest resilience.

Elephants’ spatial awareness and foraging strategies facilitate the creation of secondary habitats. For example, in Botswana’s Okavango Delta, savannah elephants strategically navigate dense thickets, opening areas that transform into grasslands, subsequently supporting herbivores like zebras and antelopes [56]. These intentional patterns of movement and resource use enhance ecosystem productivity and foster biodiversity, highlighting the cascading ecological benefits of elephant activity.

### 6.2. Opportunities to Improve Human–Elephant Interactions

#### 6.2.1. Mitigating Human–Elephant Conflict

HEC presents significant challenges for elephant conservation and the livelihoods of human communities. Understanding and leveraging elephants’ spatial memory and movement patterns can help communities in close proximity to elephant passageways design effective mitigation strategies that align with their natural behaviours [74].

Farmers have successfully reduced crop-raiding incidents in Sri Lanka’s Mahaweli region by installing beehive fences, an approach rooted in elephants’ natural avoidance of bees. These deterrents have shown up to 85% effectiveness and have been integrated into elephants’ spatial memory, prompting them to reroute movement patterns [65,68,75]. Elephants, known to avoid bees, integrate the presence of these deterrents into their mental maps, altering their movement patterns to bypass fields protected by beehives. This strategy not only safeguards crops but also promotes coexistence by addressing the root of the conflict in a manner that respects elephants’ natural behaviours and spatial awareness. Similarly, in Kenya, the implementation of wildlife corridors has been instrumental in mitigating HEC by facilitating safe passage for elephants through human-dominated landscapes. A notable example is the Mount Kenya Elephant Corridor, a 14-kilometre passage connecting the Mount Kenya Forest Reserve and NP with the Samburu lowlands. This corridor allows savannah and forest elephants to traverse these regions, reducing the likelihood of human–elephant encounters that lead to conflict.

Understanding the socio-political context of HEC is equally critical. In regions where elephants are culturally revered, communities often show higher tolerance to damage, whereas in areas where elephants are viewed as threats, conflicts escalate rapidly [76]. Compensation schemes, when timely and transparent, can foster goodwill, but if poorly implemented, they may erode trust and escalate resentment [77]. Moreover, community-based monitoring and early warning systems, such as SMS alerts or watchtowers, have proven effective in enabling proactive responses to elephant presence, especially during harvest seasons [78].

Mitigation is most successful when local knowledge and values are integrated into intervention design. For example, in northeast India, community crop guards equipped with torches, drums, and chillies patrol fields at night, techniques derived from indigenous practices and refined through experience [79]. These examples demonstrate that technical interventions alone are insufficient without parallel investments in education, community empowerment, and long-term engagement.

Initiatives that align conservation efforts with elephants’ natural migratory routes have the added benefit of promoting the well-being of elephant populations while supporting local communities’ livelihoods by decreasing crop raiding and property damage. This approach exemplifies how understanding and accommodating elephants’ spatial behaviours can lead to harmonious coexistence between humans and wildlife.

#### 6.2.2. Enhancing Community-Based Conservation

Community involvement is essential in areas where elephants and humans share landscapes. Initiatives that align conservation goals with the natural migratory patterns of elephants are particularly effective. In Namibia’s Caprivi Strip, communal conservancies have been established to designate specific corridors that elephants naturally use for migration [80]. These corridors are integrated into the community’s land-use plans, ensuring that elephants can safely navigate fragmented landscapes while minimising crop damage. Compensation programs for crop losses further incentivise local participation in conservation strategies. By aligning conservation efforts with elephants’ spatial behaviours and utilising their ranging processes to plan corridor placement, these initiatives reduce conflict and generate economic benefits through ecotourism, fostering a sense of shared stewardship between humans and elephants.

### 6.3. Conservation Strategies Informed by Elephants’ Spatial Memory and Movement Patterns

Designing wildlife corridors that align with elephants’ natural movement patterns is crucial for maintaining habitat connectivity and reducing HEC. In Kenya’s Amboseli-Tsavo ecosystem, conservationists have utilised satellite tracking data to identify and secure traditional migratory routes [81]. For instance, the successful passage of a 31-year-old male elephant named Jenga through the Amboseli-Tsavo corridor was monitored using real-time tracking technologies, ensuring safe, conflict-free movement between key habitats. This approach preserves access to seasonal resources and minimises HEC by guiding elephants along established pathways that avoid human settlements. Similarly, in India, the Kaziranga–Karbi Anglong landscape has been identified as a vital corridor supporting the seasonal migration of elephants between Kaziranga NP and the adjacent Karbi Anglong hills. However, this corridor is increasingly fragmented due to road expansion and unregulated mining activities. Studies have documented significant roadkill mortality on National Highway 715 (formerly NH-37), particularly during the monsoon migration season, prompting conservationists and the Supreme Court of India to recommend elevated corridors and underpasses to reduce wildlife–vehicle collisions [82,83,84]. In the Rajaji–Corbett corridor, Asian elephants traverse railway lines and human settlements, leading to fatal collisions. Successful mitigation strategies include early warning systems and community-based monitoring, resulting in a dramatic reduction in elephant deaths on the railway line passing through Rajaji NP [85,86]. These examples demonstrate that incorporating elephants’ spatial memory and learned movement patterns into corridor and infrastructure design enhances ecosystem connectivity and reduces human–elephant conflict.

Strategically placing artificial waterholes based on elephants’ spatial use patterns can prevent habitat degradation and promote ecological balance. Research in South Africa’s Kruger NP has shown that water provisioning influences savannah elephant spatial utilisation [87]. By analysing the distribution of water sources and elephant movements, waterholes can be positioned to minimise environmental impacts and evenly distribute grazing pressure. This strategy helps elephants integrate these resources into their spatial memory, promoting sustainable habitat use.

Developing infrastructure that accommodates elephants’ movement patterns can reduce HEC. In India’s Kaziranga NP, the construction of underpasses along major highways has significantly decreased road collisions with Asian elephants [88]. These structures align with elephants’ natural routes, allowing for safe crossings and the integration of these routes into their ranging paths. By facilitating uninterrupted movement, such infrastructure reduces the risk of accidents and fosters coexistence between elephants and human activities.

Veterinarians play an expanding role in conservation strategies that align with elephants’ spatial and cognitive behaviours. In collaring programs, veterinarians are essential not only for the safe anaesthesia and recovery of individuals but also for interpreting movement anomalies that may signal health issues. For example, collar data from Tsavo NP revealed that elephants exhibiting abrupt decreases in daily movement were often injured or nutritionally compromised, prompting on-site veterinary intervention [73]. Beyond tracking, veterinarians are critical in addressing the cognitive and physiological impacts of nutrition, particularly in captivity, where mineral imbalances can have neurological consequences. A study found that captive African elephants with suboptimal copper and selenium levels exhibited signs of reduced exploratory behaviour, impaired memory tasks, and increased stereotypic activity, suggesting a link between trace mineral deficiencies and cognitive dysfunction [89]. These findings echo research in other large herbivores, where magnesium and zinc imbalances affect synaptic plasticity and learning.

In wild populations, nutritional stress is often linked to habitat degradation and seasonal resource fluctuations. During prolonged droughts in ecosystems like the Hwange and Tarangire NPs, elephants must travel further to find forage, with reduced intake of essential nutrients such as calcium and phosphorus from depleted vegetation. Loarie et al. showed that elephants shifted from preferred grasses to mopane bark and woody browse under dry-season pressure—foods lower in digestible energy and essential minerals. These shifts not only increase gastrointestinal parasite risk but may also impair decision-making and reduce spatial precision due to chronic energy deficits [90]. Elephants under nutritional stress have been observed to abandon familiar migratory routes in search of food, exposing them to greater conflict risk and increasing mortality [91].

Veterinary nutritional strategies differ markedly between captive and wild settings. In captivity, dietary formulations can be corrected using targeted supplementation, including trace minerals and fibre balance, often monitored through blood panels and behavioural observations [89]. In contrast, veterinary teams supporting wild populations focus on habitat-level interventions—such as guiding the strategic placement of artificial waterholes or mineral licks and advising land managers on vegetation restoration to improve forage diversity. Where possible, rehabilitated elephants with known nutritional deficiencies may be transitioned using forage-rich diets to restore gut flora before reintroduction. These combined efforts show that veterinary input is indispensable for maintaining not only the physical health of elephants but also the cognitive resilience required for effective navigation, migration, and conflict avoidance.

### 6.4. Addressing Broader Conservation Goals

Elephants’ ability to remember and navigate areas allows them to play a crucial role in forest regeneration. In tropical forests like the Congo Basin, forest elephants disperse seeds of carbon-rich tree species, some of which rely exclusively on elephants for successful germination [4]. Elephants ensure genetic diversity by dispersing seeds over large areas and establishing robust forests that store large amounts of carbon. In savannah ecosystems, elephants’ browsing and trampling behaviours convert dense woodlands into open grasslands. This restructuring fosters carbon cycling and biodiversity. For example, in Botswana’s Okavango Delta, savannah elephants open thickets and create grasslands, which support grazing herbivores and promote ecosystem carbon storage [8]. Elephants contribute to ecosystem resilience by opening pathways to water sources during dry seasons, providing access for themselves and other species. In Tanzania’s Tarangire NP, this spatially informed behaviour mitigates the effects of drought on local wildlife, sustaining biodiversity [52]. Finally, elephants’ trampling behaviour mixes plant material into the soil, enhancing organic matter content. In Kenya’s Samburu region, elephants contribute to soil regeneration by dispersing seeds and depositing nutrient-rich dung, which increases soil carbon storage [65].

Elephants’ navigation and movement patterns are essential for traversing fragmented and dynamic landscapes, playing a critical role in maintaining ecosystem resilience. Their seed dispersal activities, facilitated by spatial memory and intergenerational knowledge, support forest regeneration and plant biodiversity, particularly in fragmented ecosystems like Mozambique’s Gorongosa NP [47]. Additionally, their use of waterholes, guided by detailed mental maps, provides access to critical water sources for numerous other species. By safeguarding these behaviours through protecting habitats and movement corridors, conservation efforts can maintain ecological processes crucial for adapting to rapid environmental changes.

Incorporating elephants’ spatial movement abilities into conservation policies ensures that initiatives address their behavioural ecology. For instance, the strategic placement of wildlife corridors in Kenya’s Amboseli-Tsavo ecosystem aligns with elephants’ natural routes, reducing HECs while preserving connectivity [55]. Policies prioritising intergenerational knowledge transfer, such as protecting matriarchs who hold critical spatial memory, further enhance the success of conservation strategies. Herds led by older matriarchs exhibit stronger and more immediate responses to potential dangers, underscoring the importance of the matriarch’s social memory for cohesion and survival. Long-term studies have shown that the loss of a matriarch can lead to profound disruptions in herd structure, including increased aggression, poor decision-making, and weakened social bonds [7,38,90]. Elephants orphaned or separated due to poaching or culling events display long-term deficits in social learning and movement coordination [51]. The importance of experienced elders is further underscored by historical management outcomes in Kruger NP, South Africa, where early culling programs removed only older individuals, resulting in socially disrupted and behaviourally dysregulated juvenile elephants. These “orphaned” bulls exhibited unusually aggressive behaviour, including attacks on rhinoceroses and vehicles. In response, the program was amended to remove complete family units to preserve social cohesion and reduce abnormal aggression [91].

Moreover, observations of elephants’ responses to dying or deceased matriarchs reveal complex social behaviours such as investigative touching, guarding, and vocalisations, emphasising the depth of social disruption caused by matriarch loss [92]. By designing interventions that align with elephants’ natural behaviours and movement patterns, these policies benefit elephant populations and the ecosystems they sustain, ensuring long-term ecological and social benefits [55].

## 7. Conclusions

This review underscores the pivotal role of spatial memory and movement patterns in enhancing elephants’ resilience to climate change and human–wildlife conflict. Elephants’ advanced spatial memory, supported by their sophisticated neuroanatomy, enables them to navigate fragmented habitats, locate scarce resources, and adapt their behaviours to dynamic environmental and anthropogenic challenges. While older matriarchs and experienced individuals serve as repositories of knowledge, the extent and mechanisms of knowledge transfer within herds remain largely unverified. Current evidence suggests that younger elephants learn movement patterns by following and observing older individuals, but the direct intergenerational transmission of knowledge remains speculated, with limited empirical research supporting its existence.

Understanding these spatial behaviours is crucial for designing effective conservation strategies. Protecting migratory routes and maintaining herd structures, particularly the presence of older individuals, ensures that elephants retain access to critical resources and maintain their ecological functions. Strategic wildlife corridor design, the placement of artificial waterholes, and infrastructure planning that aligns with elephants’ established movement patterns can mitigate HEC and foster coexistence. Additionally, preserving the ecological roles of elephants is essential for maintaining biodiversity and promoting climate resilience, given their influence on seed dispersal, habitat modification, and ecosystem stability. Understanding elephant spatial memory requires interdisciplinary approaches. One such methodology involves GPS-based telemetry, where high-resolution movement data from collared individuals are analysed to identify recurring routes and habitat preferences [93]. In parallel, studies on other taxa suggest that integrating such tracking data with neural mechanisms of navigation—including grid cell activity—may provide insight into the cognitive encoding of space [94]. Furthermore, the cross-generational tracking of related elephants, particularly matriarchal lines, offers evidence of culturally transmitted migratory knowledge, as seen in studies of collective animal navigation and cumulative culture [95].

Future research should focus on the neurobiological mechanisms underlying elephants’ spatial memory and how they develop and refine their movement strategies over time. Investigating the role of social influence in shaping navigation behaviours, particularly in fragmented habitats and under increasing anthropogenic pressures, could provide new insights into elephant adaptation. Moreover, integrating conservation psychology to explore human perceptions and behaviours in conflict zones may strengthen community-based conservation initiatives, ensuring sustainable and culturally appropriate strategies. Addressing HEC requires not only ecological understanding but also insights into human behaviour. Studies on community attitudes, risk perception, and tolerance levels towards elephants are essential in designing context-appropriate interventions. For example, locally adapted deterrents and land-use planning are more likely to succeed when informed by household-level interviews and participatory mapping of conflict hotspots [95,96]. Understanding why some communities tolerate crop damage while others do not can guide education, compensation schemes, and co-management strategies that are both culturally sensitive and effective in reducing conflict escalation.

By bridging behavioural ecology with conservation science, we can develop holistic solutions that enhance elephant conservation and support broader ecosystem stability and human–wildlife coexistence.

## Figures and Tables

**Figure 1 vetsci-12-00312-f001:**
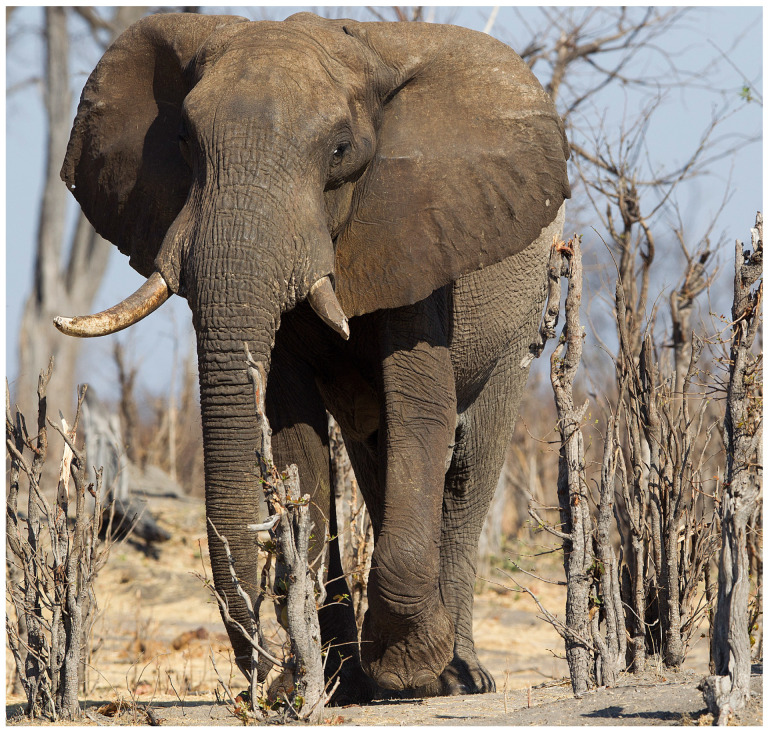
African savannah elephant [*Loxodonta africana*] (Credits: Rudi van Aarde).

**Figure 2 vetsci-12-00312-f002:**
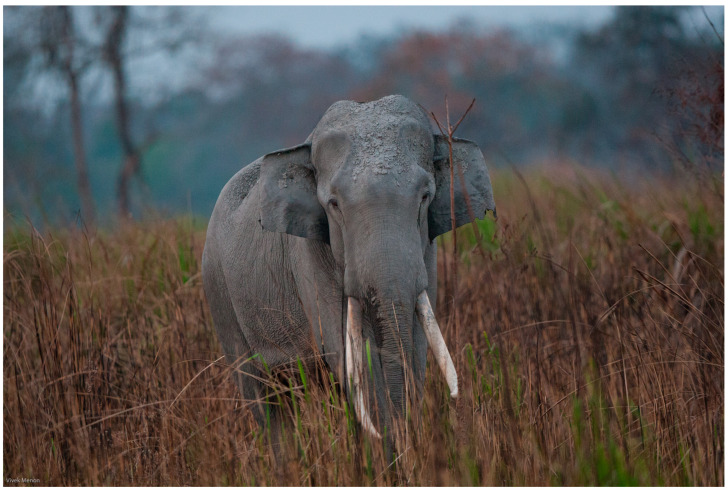
Asian elephant [*Elephas maximus*] (Credits: Vivek Menon).

**Table 1 vetsci-12-00312-t001:** Global summary of human–elephant conflict (HEC)-mitigation strategies, their implementation context, and reported effectiveness. This table synthesises key interventions used worldwide to reduce HEC, including landscape-level infrastructure, behavioural deterrents, and community-based approaches. Effectiveness varies based on ecological and socio-political context.

Mitigation Strategy	Region/Country	Description	Reported Effectiveness	Key References
Beehive fences	Kenya, Tanzania, Sri Lanka	Fences incorporating active beehives to deter elephants by exploiting their aversion to bee stings.	80–85% reduction in crop-raiding events.	King et al., 2017 [65]; Scheijen et al., 2019 [66]
Electric fencing	India, Nepal	Electrified barriers installed to exclude elephants from agricultural areas.	Variable; often bypassed or damaged; maintenance- dependent.	Sukumar, 1992 [68]
Early warning systems	India (Assam), Sri Lanka	Use of sirens, SMS alerts, and patrols triggered by elephant movement data.	Reduction in human injuries and crop loss; community- dependent.	Ahmed & Saikia, 2022 [69]
Wildlife underpasses	Kenya (Tsavo National Parks)	Engineered underpasses beneath railways or roads to facilitate safe elephant movement.	78% of elephant crossings occurred through underpasses.	Okita-Ouma et al., 2021 [70]
Chilli oil deterrent ropes	Zimbabwe, India	Ropes soaked in chilli oil strung along farm boundaries to create a chemical barrier.	Moderate success; elephants may habituate over time.	Gross et al., 2022 [71]
Wildlife corridors and land-use zoning	Namibia, Kenya	Designation and protection of natural elephant corridors across human-modified landscapes.	Effective if legally protected and community- supported.	Chase & Griffin, 2009 [72]

## Data Availability

Not applicable.

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
