# Peer review of "Memory-Based Navigation in Elephants: Implications for Survival Strategies and Conservation"

_vetsci, 2025, doi:10.3390/vetsci12040312_

Round 1
Reviewer 1 Report
Comments and Suggestions for Authors
The manuscript presents a timely and interdisciplinary synthesis of elephant spatial cognition, integrating neuroscience, behavioral ecology, and conservation policy. Its strengths lie in its comprehensive coverage of empirical and theoretical literature, actionable conservation strategies, and relevance to human-wildlife conflict mitigation. However, several issues require revision to strengthen empirical grounding, improve synthesis, and enhance clarity. Below are detailed recommendations.
Lines 2-3: Consider refining for precision, e.g., “Memory-Based Navigation in Elephants: Implications for Survival Strategies and Conservation.”
Lines 27-47: Clarify whether the paper synthesizes existing research or presents novel findings. Avoid speculative phrasing (e.g., “cognitive mapping” in Line 34) without explicit caveats.
Lines 51-56: Explicitly state the research question(s) and gaps addressed (e.g., reconciling neuroanatomical insights with field observations).
Lines 61-64: Define terminology (e.g., Euclidean maps vs. habitual route memory) to avoid ambiguity.
Lines 123-132: Differentiate between hypotheses (e.g., associative memory) and direct evidence for cognitive maps.
Lines 140-170: Link hippocampal activity to navigation behaviors through direct behavioral studies (e.g., GPS tracking correlated with neural imaging).
Lines 277-285: Discuss experimental studies (e.g., controlled trials) demonstrating knowledge transfer to younger elephants.
Lines 349-353: Specify whether observations are direct (e.g., tracking data) or inferred.
Lines 501-518: Support claims with quantitative studies (e.g., success rates of deterrents).
Lines 535-539 Include a global summary table or figure (e.g., underpasses, beehive fences).
Lines 622-627: Expand examples beyond Amboseli-Tsavo (e.g., Asian corridors like Kaziranga-Karbi Anglong).
Lines 672-675: Cite long-term studies on herd stability post-matriarch loss.
Lines 689-701: Explicitly outline methodologies (e.g., GPS-neural integration, cross-generational tracking).
Lines 702-705: Elaborate on how human behavioral studies (e.g., community attitudes) inform HEC solutions.
Minor Revisions
Line 12: Revise to “Elephants exhibit exceptional memory capabilities.”
Line 69: Specify which claims lack empirical support (e.g., “The hypothesis of Euclidean cognitive maps remains speculative due to limited experimental validation.”).
Line 166-170: Cite specific studies on neurogenesis (e.g., species, methodologies).
Please ensure consistent formatting (e.g., McComb et al. 2001 cited multiple times—condense repetitions).
Comments on the Quality of English LanguageThe quality of English is fine.
Author Response
Please see the attachement

Reviewer 2 Report
Comments and Suggestions for Authors
The authors provide an interesting review of the three species of elephant’s cognitive abilities including neural architecture and relate these to contemporary environmental challenges in their conservation. Much of the review relates to behavioural ecology, conservation science and human dimensions of wildlife management. Thus, its placement in a Veterinary journal is unusual but section 2.2 on neuroanatomy provides some justification. The review is well-written but there is tendency towards hyperbole. Thus, abilities are “extraordinary” which begs the question in a scientific discourse as to what is ordinary? Impressive navigational abilities are well-known for trans-equatorial bird migrants and amongst people, Polynesian navigation of the Pacific Ocean. Further distances and areas are “vast” when long or large would suffice. I have some more specific comments as follows:
Line 40: The authors summarise a short review of ecosystem services in 6.1 as “promoting climate resilience”. I found that overblown. The arguments were weak and hinge on seed dispersal and clearance of woody vegetation for an apparent net carbon capture. This needs more convincing data as to why elephants enhance climate resilience, but bulldozing corridors would not.
The path through section 2 about “cognitive mapping” is somewhat meandering. First no evidence but likely and later several avenues of evidence and likely.
In section 4 about the role of elders, the authors seem to be unfamiliar with extensive research in Kruger NP. A culling program initially removed old individuals (males and females) and led to “delinquent” young elephants that threatened tourists amongst various outcomes. The culling program was amended to take out family groups.
Line 381: Why is “groups” capitalised?
Line 411: “mothers” or “dams” why both?
Line 474-478: These sentences should be part of the preceding paragraph.
Line 591: incidents in Sri
Section 6.2.1. There is quite an extensive literature on mitigation and the human dimensions of human-elephant conflict. This section could be improved.
Line 660: instrumental in ranging fragmented… meaning here is unclear.
Line 709: This is confusing as it is in the singular with four authors?
Author Response
Please see attachement
